# Revisiting Multi-Codebook Quantization

## Abstract

Multi-Codebook Quantization (MCQ) is a generalized version of existing codebook-based quantizations for Approximate Nearest Neighbor (ANN) search. Therefore, MCQ theoretically has the potential to achieve the best performance because solutions of other codebook-based quantization methods are all covered by MCQ's solution space under the same codebook size setting. However, finding the optimal solution to MCQ is proved to be NP-hard due to its encoding process, *i.e.*, converting an input vector to a binary code. To tackle this, researchers apply constraints to it to find near-optimal solutions, or employ heuristic algorithms which are still time-consuming for encoding. Different from previous approaches, this paper takes the first attempt to find a deep solution to MCQ. The encoding network is designed to be as simple as possible, so the very complex encoding problem becomes simply a feed-forward. Compared with other methods on three datasets, our method shows state-of-the-art performance. Notably, our method is 11×-38× faster than heuristic algorithms for encoding, which makes it more practical for real scenery of large-scale retrieval. Our code is publicly available: `https://github.com/DeepMCQ/DeepQ`.

## 1 Introduction

Rapidly increasing multimedia contents in recent years raise an urgent request for retrieval in a short time. Unlike the exhaustive routine [31, 20], Approximate Nearest Neighbor (ANN) search significantly reduces retrieval time while preserving high recall. It has been widely applied to various scenarios, such as database indexing, fast image retrieval, and recommender systems.

As a typical approach, vector quantization (VQ) [7] is at first developed as a compression technique, which uses a codebook to approximate vectors. People further find the power of VQ to preserve similarities between quantized features and enable VQ to perform ANN search. In order to achieve low quantization errors with limited codebook size, a multi-codebook structure is introduced. The proposal of the Multi-Codebook Quantization (MCQ) [2] describes the approach as a combination of one codeword for each sub-codebook, and previous methods [9, 6, 19, 30, 10, 3] are summarized as exceptional cases of MCQ or *constrained MCQ*s. The quantization codes are designed to be compacted, which results in negligible storage cost and high-quality results.

However, the optimization of MCQ without any constraints is formally NP-hard. [14] models it as the minimization on several fully-connected Markov Random Fields (MRFs). As a result, current researches aim at solving MCQ under acceptable computational costs. Other than applying constraints on it [34, 4, 15], another approach designs algorithms in a heuristic way [2, 14, 16]. The latter achieves better performance but suffers from slow encoding.

There are chances to employ neural networks' power to solve MCQ, where people expect to obtain higher performance and encoding efficiency than previous methods. [11, 5, 28, 33, 27] already give the way to treat codebook as network parameter and update it by gradient-descent, but they are

all still under constraints that hinder performance. Morozov and Babenko [18] and Sablayrolles *et al.* [22] map datapoints to learned space, which are not flexible, especially when performing the reconstruction. Therefore in this paper, we give our first attempt to solve MCQ in a deep learning approach, without constraints and work-arounds. Our contributions can be summarized as three-folds:

- Our novel approach, Deep Multi-Codebook Quantization (DeepQ), fully considers encoding difficulty and time complexity in MCQ. With the high efficient and parallelized encoding networks, our method significantly reduces encoding time.

- To tackle the NP-hard encoding problem and non-differentiable gradient estimation, we employ and further revise a policy gradient method. Value-Corrected Proximal Policy Optimization (VC-PPO) is proposed to speed up convergence in the training phase.

- Experiments conducted on a benchmark dataset validate our proposed method. Furthermore, to evaluate the scalability of the method, it is tested on million-scale datasets to show the effectiveness of our proposed algorithm.

## 2 Related Works

Vector quantization is a routine to approximate vectors by a codebook. Typical applications include clustering, compression, and Approximate Nearest Neighbor (ANN) search. The famous proposal $k$-means [7], also known as Lloyd's algorithm [13], clusters the dataset into uniformly sized convex cells. When it is applied to ANN search, datapoints from the base set are quantized into their nearest centriods and represented by indices. The distance from a given query to any datapoint is approximated by the distance from the query to the datapoint's centriod, which is effectively pre-computed and stored in a lookup table. To perform fine-grained clustering as well as reducing the space and time complexity, they [9, 6, 19, 10, 30] divide the feature space orthogonally by performing $k$-means in each subspace concurrently. Meanwhile, the introduced sub-codebook structure reveals the prototype of MCQ. Formally, [2] gives a well definition of MCQ, and previous works are all summarized into constrained MCQs. Specifically, subspace $k$-means must keep orthogonality among sub-codebooks. Zhang *et al.* [34] loosens the orthogonality constraint, but sub-codebooks are still weakly-orthogonal. Chen *et al.* [4] and Martinez *et al.* [15] propose hierarchical $k$-means, where vectors are quantized coarse-to-fine. If constraints are moved, MCQ is not easy to solve. Current state-of-the-art methods develop heuristic algorithms to help to encode. Specifically, Babenko and Lempitsky [2] employs beam search, Martinez *et al.* [14, 16] give algorithm based on Iterated Conditional Modes (ICM). However, the above methods do not achieve satisfied time complexity in encoding yet.

When neural networks and gradient descent become a fashion, a few attempts to integrate quantization into deep retrieval networks are proposed. Klein and Wolf [11] and Song *et al.* [5] propose Deep Product Quantization (DPQ) and Deep Progressive Quantization (DPgQ) which update codebook by soft relaxation, but they are still under the same constraints as [9, 15]. Sablayrolles *et al.* [22] and Morozov and Babenko [18] give pipelines to encode compact representations for compressed-domain search, but they do not strictly follow the paradigm of MCQ.

## 3 Preliminaries

Given a vector $\boldsymbol{x} \in \mathbb{R}^D$, its quantized vector $\tilde{\boldsymbol{x}}$ are composed by several codewords in a codebook $\boldsymbol{C}$. More Specifically, $\boldsymbol{C} = (\boldsymbol{C}_m)$, $\boldsymbol{C}_m \in \mathbb{R}^{K \times D}$, $1 \le m \le M$ contains $M$ sub-codebooks and $K$ codewords for each. Quantization codes are formed by $\boldsymbol{b} = (\boldsymbol{b}_m)$, $\boldsymbol{b}_m \in \{1, 2, \cdots, K\}$, $1 \le m \le M$, which indicates the picked codeword in each sub-codebook. For the whole training set $\boldsymbol{X} = \{\boldsymbol{x}\}$ with $N$ datapoints, MCQ aims at finding the optimal quantization codes $\boldsymbol{B} = \{\boldsymbol{b}\}$ and codebook $\boldsymbol{C}$ to minimize following objective:

$$\min_{\boldsymbol{C},\boldsymbol{B}} \mathbb{E}_{\substack{\boldsymbol{x} \in \boldsymbol{X} \\ \boldsymbol{b} \in \boldsymbol{B}}} \mathrm{Q}\left(\boldsymbol{x}, \boldsymbol{b}, \boldsymbol{C}\right) = \min_{\boldsymbol{C},\boldsymbol{B}} \mathbb{E}_{\substack{\boldsymbol{x} \in \boldsymbol{X} \\ \boldsymbol{b} \in \boldsymbol{B}}} \left\| \boldsymbol{x} - \sum_{m=1}^{M} \boldsymbol{C}_{m\boldsymbol{b}_m} \right\|_2 \tag{1}$$

where $\boldsymbol{C}_{m\boldsymbol{b}_m} \in \mathbb{R}^D$ is the $\boldsymbol{b}_m$-th codeword of the $m$-th sub-codebook. The sum of picked codewords $\sum \boldsymbol{C}_{m\boldsymbol{b}_m}$ tries to approximate $\boldsymbol{x}$. $\boldsymbol{C}$ and $\boldsymbol{b}$ are stored for further retrieval. Some of the previously mentioned methods [9, 6, 4, 15, 34] are treated as *constrained MCQ*s, as they are all represented

as special cases of (1). Specifically, when $M = 1$, (1) becomes VQ. Or if any two sub-codebooks $C_i, C_j$ are orthogonal, it will be PQ or OPQ.

The optimization of (1) without any constraints is proved to be NP-hard [14]. To tackle this, we propose a Expectation-Maximization style solution. Following sections will explain the deep neural network for encoding $b$ (Section 4.1), the way to solve $C$ (Section 4.2), and how to conduct retrieval (Section 4.3), respectively.

## 4 Methodology

### 4.1 Expectation: Encoding $B$ with neural networks

Our first step, is to find a potential code $b$ by given $x$ and a fixed $C$. A policy $\pi$ parameterized by $\theta$ is employed to take possible solution of $b$ by feeding $x$:

$$\pi = (\pi_m) = \pi\left(x \mid \theta_m\right), \ 1 \le m \le M. \tag{2}$$

More specifically, $\pi$ produces $M$ Categorical distributions $\text{Categorical}(K, p_{m1}, \cdots, p_{mK})$, where $p_{mj}$ is the probability to pick the $j$-th codeword in the $m$-th sub-codebook. A potential encoding $b_m$ is generated by drawing samples from $\pi_m$, which then helps us to pick codeword $C_{mb_m}$. Therefore:

$$b_m \sim \pi_m\left(x \mid \theta_m\right) = \text{Categorical}(K, p_m). \tag{3}$$

Since the independence among different sub-codebooks is a prerequisite of MCQ, $b_m$ should be drawn from $\pi_m$ *independently*. Intuitively, the probability of $b$ to be a specific $b^\star$ is derived by conditional independence:

$$\Pr\left(b = b^\star\right) = \prod_{m=1}^{M} \Pr\left(b_m = b_m^\star\right) = \prod_{m=1}^{M} p_{mb_m^\star}. \tag{4}$$

We adopt the power of neural networks to model $\pi_m$. Specifically, $\theta_m$ produces $K$ unnormalized log-probabilities $\ell_m$ and $p_{mj}$ is obtained by Softmax. To keep the independence, $\theta_m$ will not share parameters with each other.

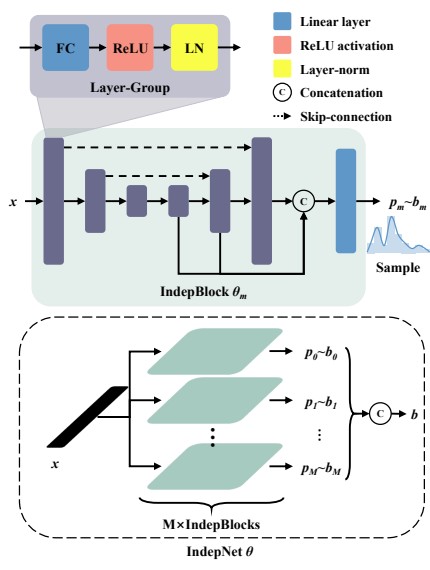

Figure 1: Our proposed *IndepNet* for producing probabilities of choosing each codeword. *IndepBlock* is duplicated for $M$ times without shared parameters, in order to keep independence between different IndepBlocks. Categorical distribution is built upon output from the Indep-Block. Then, quantization code $b_m$ associated with sub-codebook $C_m$ is sampled from distribution.

Therefore, $\theta$, or our proposed *IndepNet* is illustrated in Figure 1. We first build a basic structure called *IndepBlock* and duplicate this block for $M$ times as $\theta_1, \theta_2, \cdots, \theta_M$. We try to keep the basic structure really simple to achieve high efficiency during training and encoding. As the figure shows, IndepBlock is an hourglass network contains 6 layer-groups (consists of a linear layer with ReLU activation and layer-normalization) with skip-connections. The last three outputs are concatenated and further fed into a final linear layer with $K$ outputs as $\ell_m = (\ell_{m1}, \cdots, \ell_{mK})$, and therefore:

$$p_{mj} = \text{Softmax}\left(\ell_m\right)_j, \ where \ \ell_m = \theta_m\left(x\right). \tag{5}$$

#### 4.1.1 Gradient estimation

The objective of training $\theta$ is formed as:

$$\min_{\pi} \ \underset{\substack{x \in X \\ b \sim \pi(x|\theta)}}{\mathbb{E}} \ Q\left(x, b, C\right). \tag{6}$$

However, the optimization faces two problems: 1) The encoding of $b$ involves sampling from discrete distributions, which is non-differentiable, 2) All possible encoding of $b$ is $\mathcal{O}\left(K^M\right)$. Exhaustive search becomes impracticable.

Therefore, gradient estimation over discrete, stochastic computation graph is required to train $\theta$. Mainstream methods [23, 32, 17] include score function gradient estimator, pathwise gradient estimator, *etc*. Meanwhile, minimizing (6) is also faced with the high-variance problem during gradient estimation. To tackle this, the advantage function is introduced [12, 25]. Specifically in our work, a value network called *QENet* parameterized by $\tau$ is proposed to model a value function $v = \mathrm{V}\left(\cdot \mid \tau\right)$. It performs a regression task to minimize the following objectives:

$$\min_\tau \ \mathop{\mathbb{E}}_{\substack{\boldsymbol{x} \in \boldsymbol{X} \\ \boldsymbol{b} \sim \pi(\boldsymbol{x}|\theta)}} \ \|\mathrm{Q}\left(\boldsymbol{x}, \boldsymbol{b}, \boldsymbol{C}\right) - \mathrm{V}\left(\boldsymbol{x}, \boldsymbol{b}, \boldsymbol{C} \mid \tau\right)\|_2. \tag{7}$$

Advantages $\hat{A}$ is then estimated by

$$\hat{A} = \mathrm{Q}\left(\boldsymbol{x}, \boldsymbol{b}, \boldsymbol{C}\right) - \mathrm{V}\left(\boldsymbol{x}, \boldsymbol{b}, \boldsymbol{C} \mid \tau\right). \tag{8}$$

The detailed architecture of *QENet* is shown in Figure 2. We reuse the IndepBlock to generate $v$ by $M + 1$ blocks: $\tau = (\tau_1, \cdots, \tau_M, \tau_{\boldsymbol{x}})$. Specifically, latent representation for each selected-codeword $\boldsymbol{C}_{m\boldsymbol{b}_m}$ is obtained by:

$$\boldsymbol{\iota}_m = \tau_m(\boldsymbol{C}_{m\boldsymbol{b}_m}). \tag{9}$$

The last IndepBlock $\tau_{\boldsymbol{x}}$ is introduced to transform $\boldsymbol{x}$. Then, all the outputs from IndepBlocks are summed up to get scalar value $v$ (denoted as "reduce-sum"):

$$v = \mathrm{sum}(\boldsymbol{\iota}_1, \cdots, \boldsymbol{\iota}_M, \boldsymbol{\iota}_x). \tag{10}$$

**Value-corrected proximal policy optimization** We propose a variant of score function gradient estimator called Value Corrected Proximal Policy Optimization (VC-PPO) based on PPO to get simple but efficient Trust Region updates [26, 24]. In the real scenario of large-scale

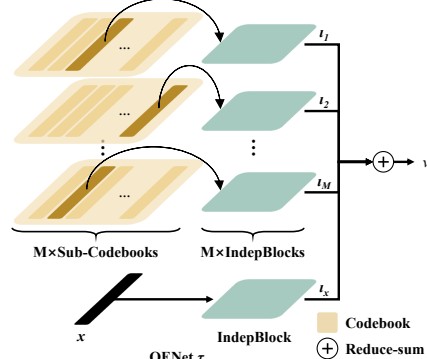

Figure 2: Our proposed *QENet* for advantage estimation. First $M$ Indep-Blocks are fed by $M$ selected codewords and the last one is fed by $\boldsymbol{x}$. Outputs are summed up to get scalar value $v$.

ANN search, the training size $N$ is usually larger than $10k$. Conventional PPO still does not satisfy us due to the speed of convergence. Therefore, we revise and propose the Value-Corrected PPO (VC-PPO) to achieve fast training. Firstly in the sampling stage, $\boldsymbol{b}_o$ and $v_o$ is produced from datapoint $\boldsymbol{x}$ over whole training set $\boldsymbol{X}$ by freezing current policy network and value network as $\theta_o, \tau_o$:

$$\begin{aligned} \boldsymbol{b}_o &\sim \pi\left(\boldsymbol{x} \mid \theta_o\right), \\ v_o &= \mathrm{V}\left(\boldsymbol{x}, \boldsymbol{b}_o, \boldsymbol{C} \mid \tau_o\right). \end{aligned} \tag{11}$$

The probability of producing the sampled $\boldsymbol{b}_o$ is denoted as $p_o = \mathrm{Pr}\left(\boldsymbol{b}_o \mid \boldsymbol{x}, \theta_o\right)$, calculated by equation (4). Finally, our surrogate objectives of VC-PPO is defined as [8]:

$$\begin{aligned} \mathcal{L}_\theta = \min \Bigg( &\frac{\mathrm{Pr}\left(\boldsymbol{b}_o \mid \boldsymbol{x}, \theta\right)}{\mathrm{Pr}\left(\boldsymbol{b}_o \mid \boldsymbol{x}, \theta_o\right)} \hat{A}, \\ &\mathrm{clip}_{1-\epsilon}^{1+\epsilon}\left(\frac{\mathrm{Pr}\left(\boldsymbol{b}_o \mid \boldsymbol{x}, \theta\right)}{\mathrm{Pr}\left(\boldsymbol{b}_o \mid \boldsymbol{x}, \theta_o\right)}\right)\hat{A}\Bigg), \end{aligned} \tag{12}$$

$$\mathcal{L}_\tau = \max \Big( \left(\mathrm{Q}\left(\boldsymbol{x}, \boldsymbol{b}_o, \boldsymbol{C}\right) - \mathrm{V}\left(\boldsymbol{x}, \boldsymbol{b}_o, \boldsymbol{C} \mid \tau\right)\right)^2, \\ \left(\mathrm{Q}\left(\boldsymbol{x}, \boldsymbol{b}_o, \boldsymbol{C}\right) - v_o - \mathrm{clip}_{-\epsilon}^{+\epsilon}\left(\mathrm{V}\left(\boldsymbol{x}, \boldsymbol{b}_o, \boldsymbol{C} \mid \tau\right) - v_o\right)\right)^2\Big). \tag{13}$$

Here, The $\mathrm{clip}\left(\cdot\right)$ forces the policy and value to be not too far from old ones and $\epsilon$ is the clip-range. In both equations, it prevents a large update ratio leading to an unstable policy. The key difference between the original PPO and our VC-PPO is, we use $\mathrm{V}\left(\boldsymbol{x}, \boldsymbol{b}_o, \boldsymbol{C} \mid \tau\right)$ other than the recorded old value $v_o$ from sampling stage to estimate advantage. This modification is treated as a value-correction process. Correcting value leads to a precise estimation on advantage, which is based on two reasons: a) Biases are introduced into advantage estimation if we use $v_o$, since the policy is getting better and better during training but $\tau_o$ is froze, and 2) The calculation of $\mathrm{V}\left(\boldsymbol{x}, \boldsymbol{b}_o, \boldsymbol{C} \mid \tau\right)$ can be done instantly

without introducing significant computational overhead. To further encourage the network choose codewords uniformly, a regularization is applied to $\theta$ to maximize the entropy of $\pi$:

$$e_\theta = - \sum_{m=1}^{M} \sum_{j=1}^{K} \boldsymbol{p}_{mj} \log \boldsymbol{p}_{mj} \tag{14}$$

which forces network to try more codeword combinations.

### 4.2 Maximization: Solve $C$ by least-squares

To give the closed-form derivation of solving $C$ by given $X$ and $B$, We will firstly rewrite Equation (1) to a matrix formulation. Since $\boldsymbol{b} = (\boldsymbol{b}_1, \boldsymbol{b}_2, \cdots, \boldsymbol{b}_M)$ and $\boldsymbol{b}_m \in \{1, 2, \cdots K\}$ is the index of selected codeword in the $i$-th sub-codebook, a one-hot encoding and a concatenation on each $\boldsymbol{b}_m$: $\boldsymbol{b}'_m = \text{one-hot}(\boldsymbol{b}_m)$, $\boldsymbol{b}' = (\boldsymbol{b}'_1, \cdots, \boldsymbol{b}'_m)$ will convert the quantization code to a $M$-hot vector *i.e.* a vector that contains $M$ segments, and each segment contains exactly one 1 and remaining 0, where 1 is the entry of picked codeword. Correspondingly, a reshape is applied to $C$: $\boldsymbol{C}' = \begin{pmatrix} \boldsymbol{C}_1 \\ \boldsymbol{C}_2 \\ \vdots \\ \boldsymbol{C}_M \end{pmatrix} \in$

$\mathbb{R}^{(M \times K) \times D}$. (1) will become:

$$\min_{\boldsymbol{C}'} \left\| \boldsymbol{X} - \boldsymbol{B}' \boldsymbol{C}' \right\|_2^2. \tag{15}$$

This equation is formally a linear least-squares regression, where $\boldsymbol{B}' \in \{0, 1\}^{N \times (M \times K)}$ is known and $\boldsymbol{X}$ is target. Although there is a bunch of algorithms to solve it, we finally choose *gelsy* [1], which in our experiments shows the best results. The solution is to first apply a QR factorization with column permutation on $\boldsymbol{B}'$ :

$$\boldsymbol{B}' = \boldsymbol{Q} \begin{pmatrix} \boldsymbol{R}_{11} & \boldsymbol{R}_{12} \\ 0 & \boldsymbol{R}_{22} \end{pmatrix} \boldsymbol{P}^\intercal \tag{16}$$

where $\boldsymbol{Q}$ and $\boldsymbol{R} = \begin{pmatrix} \boldsymbol{R}_{11} & \boldsymbol{R}_{12} \\ 0 & \boldsymbol{R}_{22} \end{pmatrix}$ is the factorization matrix and $\boldsymbol{P}$ is an orthogonal matrix that permutes columns of $\boldsymbol{B}'$ until $\boldsymbol{R}_{11}$ is well-conditioned (its estimated condition number approaches 0). With the permutation, $\boldsymbol{R}_{22}$ becomes negligible. Moreover, $\boldsymbol{R}_{12}$ is erased by another orthogonal transformation:

$$\begin{pmatrix} \boldsymbol{R}_{11} & \boldsymbol{R}_{12} \\ 0 & \boldsymbol{R}_{22} \end{pmatrix} \rightarrow \begin{pmatrix} \boldsymbol{R}_{11} & \boldsymbol{R}_{12} \\ 0 & 0 \end{pmatrix} = \begin{pmatrix} \boldsymbol{T}_{11} & 0 \\ 0 & 0 \end{pmatrix} \boldsymbol{Z} \tag{17}$$

where $\boldsymbol{T}$ and $\boldsymbol{Z}$ are from the orthogonal transformation of $\boldsymbol{R}$. Then, $\boldsymbol{C}'$ is derived by:

$$\boldsymbol{B}' = \boldsymbol{Q} \begin{pmatrix} \boldsymbol{T}_{11} & 0 \\ 0 & 0 \end{pmatrix} \boldsymbol{Z} \boldsymbol{P}^\intercal,$$
$$\boldsymbol{C} \leftarrow \boldsymbol{C}' \leftarrow \boldsymbol{P} \boldsymbol{Z}^\intercal \begin{pmatrix} \boldsymbol{T}_{11}^{-1} \boldsymbol{Q}_1^\intercal \boldsymbol{X} \\ 0 \end{pmatrix} \tag{18}$$

where $\boldsymbol{Q}_1$ is the top $\text{rank}(\boldsymbol{B}')$ columns of $\boldsymbol{Q}$.

In brief, our overall training approach is summarized into algorithm 1.

### 4.3 Fast retrieval

After training, we are able to encode the base set for retrieval. Other than sampling from $\pi$, codewords are simply rolled out by greedy assignments:

$$\boldsymbol{b}_m^g = \arg \max \theta_m(\boldsymbol{x}). \tag{19}$$

We firstly use the greedy roll-out strategy to obtain $\boldsymbol{B}$ in the training set in order to solve the final codebook. Then, we employ the same strategy to encode the base set.

To further refine assignments, we add an extra step that randomly selects and alters $\boldsymbol{b}_i$ while fixing others:

$$\boldsymbol{b}_i^g \leftarrow \underset{\boldsymbol{b}_i^g}{\arg\min} \, \mathrm{Q}\left(\boldsymbol{x}, \boldsymbol{b}^g, \boldsymbol{C}\right),$$
$$i \sim \mathcal{U}[1, M]. \tag{20}$$

Since this refinement only causes negligible overhead referred to the implementation by [14], in practice, we benefit from it not only to get lower quantization error but also to obtain acceptable performance from a fast training, *i.e.*, training within a very few steps before the network is converged.

The encoded and refined base set, combined with the codebook, is finally employed for retrieval. The LSQ-style lookup table [14] is utilized to speed up similarity search.

### 4.4 Discussion

Our work aims at solving Multi-Codebook Quantization via neural networks. Similar works include Unsupervised Neural Quantization (**UNQ**) [18] and **Spreading Vectors** [22]. But ours has several key advantages compared to previous works: **1)** Unlike UNQ, which reconstructs features by an encoder-decoder structure, we follow the paradigm of MCQ to directly give binary codes and codebooks for the benefit of speed and storage, for UNQ needs an extra decoding stage during retrieval. **2)** UNQ and Spreading Vectors both project original features into a learned space. Although similarities between features are preserved, they still have biases in quantized results. This causes several issues, especially when we want to perform a reconstruction to approximate original features, *e.g.* data compression.

Compared to LSQ [14], the state-of-the-art heuristic algorithm, our work is the first to tackle MCQ in a deep learning fashion. The policy network is designed to be very simple to get fast encoding speed and comparable retrieval performance.

---

**Algorithm 1:** VC-PPO for Training

**Inputs:** Training set $\boldsymbol{X}$, max step $T$, hyper-parameters $\alpha$, $\epsilon$, learning rates $\eta_1$, $\eta_2$.
**Outputs:** Policy $\pi$.
Initialize codebook $\boldsymbol{C}$, parameters $\theta$ and $\tau$;
$i \leftarrow 0$;
**while** $i < T$ **do**
    /* Training loop */
    **for** $\boldsymbol{x}$ *in* $\boldsymbol{X}$ **do**
        /* Sampling stage */
        Sample $\boldsymbol{b}_o \sim \pi\left(\boldsymbol{x} \mid \theta_o\right)$ into $\boldsymbol{B}$;
        Compute $v_o, p_o$ into $\boldsymbol{V}, \boldsymbol{Pr}$;
    **end**
    **for** $\boldsymbol{x}, \boldsymbol{b}_o, v_o, p_o$ *in* $\boldsymbol{X}, \boldsymbol{B}, \boldsymbol{V}, \boldsymbol{Pr}$ **do**
        /* Updating stage */
        $\tau \leftarrow \tau - \eta_1 \nabla_\tau \mathcal{L}_\tau$;
        Compute $\hat{A}$ by (8);
        $\theta \leftarrow \theta + \eta_2 \nabla_\theta (\mathcal{L}_\theta + \alpha \cdot e_\theta)$;
    **end**
    $\boldsymbol{C} \leftarrow$ Solved by (15) ~ (18);
    $i \leftarrow i + 1$;
**end**
**return** $\pi\left(\cdot \mid \theta\right)$

---

## 5 Experiments

Our proposed Deep Multi-Codebook Quantization (**DeepQ**) is compared against the state-of-the-arts on a visual-feature dataset (**LabelMe22K**) to evaluate retrieval performance and encoding speed. Then, we scale up to make comparisons on commonly used large-scale datasets (**SIFT1M** and **DEEP1M**), whose base sets include 1 million vectors for retrieval. Furthermore, ablation study on **SIFT1M** investigates the effectiveness of each component in our proposed pipeline.

### 5.1 Datasets and evaluation metrics

**LabelMe22K** [29]: This dataset collects images by the LabelMe annotation tool[1] and uses Convolutional Neural Network (CNN) to extract them into 512-d features. It has $22,019$ vectors for training and $2,000$ vectors for test.

**SIFT1M**[2] and **DEEP1M**[3]: Both datasets contain $10^4$, $10^5$, $10^6$ vectors in query, training and base set, respectively. Vectors from SIFT1M is extracted by Scale-Invariant Feature Transform (128-d) while DEEP1M contains 96-d vectors from outputs of a CNN.

---

[1]https://github.com/CSAILVision/LabelMeAnnotationTool
[2]http://corpus-texmex.irisa.fr/
[3]http://sites.skoltech.ru/compvision/noimi/

226 Recall@$\{1, 10, 100\}$ and quantization error are adopted as evaluation metrics. These two metrics
227 indicate not only the retrieval performance but also the reconstruction accuracy. Because LabelMe22K
228 does not have a base set, its training set is adopted as a base set. We train on the training set, and then
229 encode the base set for evaluations with queries. When calculating recall, groundtruth is defined as
230 the nearest neighbor of each query in the base set (sorted by $l2$ distance). As for quantization error,
231 the average value of $Q(\boldsymbol{x}, \boldsymbol{b}, C)$ is reported over all $\boldsymbol{x}$ in the base set.

232 We compare our proposal with both shallow and deep methods, including three classic quantiza-
233 tion: **OPQ** [6], **SQ** [15] and **LSQ++** [14, 16] (denoted as **LSQ** for simplicity. Also, these two in
234 our experiments have similar performance), as well as three gradient-based methods: **DPQ** [11],
235 **DPgQ** [5] and **DRQ** [28]. **DPQ** and **PQNet** [33] have basically the same architecture that extend
236 **PQ** with gradient-descent, so we only report the performance of **DPQ**. Additionally, **UNQ** [18] is
237 also included, although they introduce an extra decoder and re-ranking trick for retrieval.

## 5.2 Implementation details

239 Our method is implemented with Py-
240 Torch,[4] the popular deep learning
241 package in Python. Codebook $C$ is
242 solved by Intel MKL that has been
243 fully optimized for speed. As for net-
244 work training, we adopt Adam opti-
245 mizer with AMSGrad [21] and hyper-
246 parameters are tuned by grid search.
247 Specifically, learning rates $\eta_1 = \eta_2 =$
248 $2 \times 10^{-4}$, with an exponetial learning
249 rate decay $\gamma = 0.9999$. Batch-size
250 in updating stage is 2000, while other
251 hyper-parameters $\epsilon = 0.2$, $\alpha = 0.05$.
252 Additionally, during training, we in-
253 sert dropout layers after every layer-
254 normalization in all layer-groups to
255 tackle overfitting. More detailed set-
256 tings as well as specifications of *In-*
257 *depNet* $\theta$ and *QENet* $\tau$ on each dataset

| Method | LabelMe22K | | | | | |
| | 32 bits | | | 64 bits | | |
| | R@1 | R@10 | R@100 | R@1 | R@10 | R@100 |
| --- | --- | --- | --- | --- | --- | --- |
| OPQ | 18.70 | 57.25 | 90.10 | 32.30 | 80.40 | 98.00 |
| SQ | 18.45 | 57.60 | 90.85 | 32.65 | 82.05 | 99.05 |
| LSQ | 21.20 | 60.85 | 94.35 | 36.45 | 86.25 | 99.15 |
| DPQ | 8.60 | 32.80 | 77.50 | 15.35 | 48.75 | 90.75 |
| DPgQ | 19.85 | 57.80 | 90.70 | 35.05 | 84.10 | 98.90 |
| DRQ | 9.65 | 34.15 | 80.15 | 30.75 | 77.35 | 97.10 |
| UNQ | 22.25 | 61.20 | 89.30 | 37.10 | 85.55 | 98.80 |
| Ours | **24.45** | **69.05** | **97.65** | **39.60** | **87.60** | **99.80** |

Table 1: Recall(R)@$\{1, 10, 100\}$ on **LabelMe22K** dataset (%). Ours outperforms state-of-the-arts by at least 2.20%, 7.85%, 3.30% (32 bits), and 2.70%, 1.45%, 0.65% (64 bits), respectively.

258 (LabelMe22K, SIFT1M, DEEP1M) can be found in supplementary material.

259 As for quantization code-lengths, $K = 256$ codewords for each sub-codebook and $M = \{4, 8\}$
260 sub-codebooks are employed in total. We follow [2] to report "effective" code-lengths (additional
261 code-length for storing $\|\boldsymbol{x}\|$ for lookup table is ignored). Therefore code-lengths become $\{32, 64\}$
262 bits, respectively.

263 For a fair comparison, experiments are conducted on a single machine, equipped with Intel Xeon
264 E5-2678v3 CPU, 256 GiB RAM, and NVIDIA RTX 3090 GPU. For other methods, we re-run on all
265 datasets under unified settings with implementations provided by the authors.

## 5.3 Comparisons with state-of-the-arts

267 Under the small training set and base set settings on LabelMe22K, we get the results placed in
268 Table 1. Our method takes the highest recall on this dataset, outperforming the state-of-the-art
269 by 2.20%, 7.85%, 3.30% on 32 bits for R@1, R@10 and R@100. It also outperforms the best
270 competitor by 2.70%, 1.45%, 0.65% on 64 bits. In brief, All methods except for UNQ are generally
271 split into three styles: **1)** *PQ-like*: OPQ and DPQ. **2)** *SQ-like*: SQ, DPgQ and DRQ. **3)** *MCQ*: LSQ
272 and ours. Generally, DPQ, DPgQ, and DRQ achieve similar results compared to their shallow
273 versions. However, since they are still constrained MCQs, they show worse performances than 3).
274 The performance of LSQ is worse than ours, shows the effectiveness of neural networks for modeling
275 the MCQ encoding problem. As for UNQ, it takes several extra tricks *i.e.*, another network for
276 decoding and re-ranking in retrieval. Although it beats LSQ, our network still shows the power of
277 MCQ to win the competition.

---

[4]https://pytorch.org/

| Method | SIFT1M | | | | | | DEEP1M | | | | | |
|--------|--------|--------|---------|--------|--------|---------|--------|--------|---------|--------|--------|---------|
| | 32 bits | | | 64 bits | | | 32 bits | | | 64 bits | | |
| | R@1 | R@10 | R@100 | R@1 | R@10 | R@100 | R@1 | R@10 | R@100 | R@1 | R@10 | R@100 |
| OPQ | 5.34 | 22.03 | 56.72 | 22.84 | 60.27 | 92.19 | 3.07 | 15.39 | 48.40 | 15.34 | 50.06 | 87.96 |
| SQ | 9.45 | 34.88 | 70.07 | 24.41 | 65.48 | 93.17 | 6.41 | 26.79 | 70.25 | 19.95 | 56.31 | 91.27 |
| LSQ | **11.43** | **40.48** | **80.52** | **33.23** | **78.37** | **98.72** | 7.29 | 28.96 | **72.93** | **21.12** | 61.47 | 93.98 |
| DPQ | 5.41 | 22.97 | 58.57 | 21.87 | 59.39 | 91.66 | 1.59 | 8.96 | 33.09 | 9.53 | 33.45 | 72.80 |
| DPgQ | 9.71 | 35.03 | 74.19 | 27.96 | 69.98 | 96.04 | 6.36 | 26.16 | 70.02 | 18.98 | 55.80 | 90.95 |
| DRQ | 1.40 | 8.87 | 35.27 | 18.56 | 53.06 | 88.45 | 4.48 | 22.46 | 62.57 | 16.10 | 52.76 | 89.31 |
| UNQ | 10.01 | 33.92 | 73.39 | 28.37 | 69.15 | 95.99 | 5.19 | 23.55 | 65.09 | 16.12 | 52.06 | 90.10 |
| Ours | 11.02 | 37.73 | 76.79 | 28.02 | 70.22 | 96.43 | **7.43** | **30.03** | 72.48 | 20.87 | **62.06** | **94.07** |

Table 2: Quantitative comparisons with state-of-the-arts on **SIFT1M** and **DEEP1M** datasets. Recall(R)@$\{1, 10, 100\}$ are reported (%). Ours shows comparable performance with staet-of-the-arts on SIFT1M, while achieving the highest recall in most cases on DEEP1M.

### 5.3.1 Large-scale retrieval performance

Our evaluations on SIFT1M and DEEP1M datasets is presented in Table 2. The training set and base set are scaled up, and retrievals on these datasets become more difficult. We observe expected results on two datasets. Compared to our main competitor, LSQ, our method achieves comparable performance on SIFT1M, and outperforms LSQ on DEEP1M in most cases. Our method achieves higher recall on DEEP1M than SIFT1M. A potential reason is that DEEP1M is under a nearly normal distribution that, in practice, is easier to converge than SIFT1M, which has a larger variance between datapoints. The performance of UNQ in our experiments is lower than expected, possibly due to different dataset settings.

Another key advantage of our method is that, different from shallow methods, which are hand-crafted algorithms that find possible solutions manually or with constraints, our DeepQ encodes vectors by only a feed-forward.

### 5.3.2 Encoding efficiency

In order to verify the encoding efficiency of our method, evaluations of encoding time on SIFT1M with the $10^6$ base set are conducted by checking the total time spent. All of them are run under GPU-acceleration. Additionally, we evaluate the time with and without the extra codewords refinement that introduced in section 4.3 (128 bits results are simulated). As Figure 3 shows, our network is significantly faster than LSQ since it needs to perform local search iteratively for 25 or even 100 rounds. Specifically, to encode SIFT1M base set, LSQ takes 52.84s, 96.99s, 256.86s and 639.18s for 16, 32, 64 and 128 bits respectively. By contrast, our method takes 4.46s, 5.46s, 8.26s and 16.64s, which is $11.8\times$, $17.8\times$, $31.1\times$ and $38.4\times$ faster than LSQ. Moreover, our method is even faster than most of the constrained MCQs. We also notice that the refinement takes negligible overhead. Although UNQ takes the fastest encoding speed, it still needs to decode and re-rank during retrieval, which slows down its retrieval speed.

### 5.3.3 Reconstruction accuracy

The comparisons of quantization error on three datasets are stated in Table 3. Basically, when the quantization error gets lower, recall will be higher.

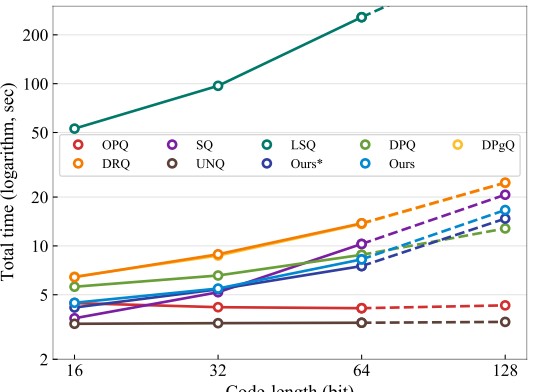

Figure 3: Total encoding time w.r.t. code-length on SIFT1M dataset. For 128 bits, we illustrate the simulated results. The variant *Ours\** removes extra refinement step to show its overhead. Our two variants are significantly faster than LSQ while achieving similar performance. Furthermore, our method is slightly faster than most of the constrained MCQs. Our method achieves high performance as well as superior encoding efficiency. UNQ has the shortest time to encode the whole set, however during retrieval, they still need to decode and re-rank that slow down the speed.

| Method | SIFT1M | | DEEP1M | | LabelMe22K | |
|---|---|---|---|---|---|---|
| | 32 bits | 64 bits | 32 bits | 64 bits | 32 bits | 64 bits |
| OPQ | $4.03 \times 10^4$ | $2.51 \times 10^4$ | $4.25 \times 10^{-1}$ | $2.70 \times 10^{-1}$ | $1.25 \times 10^{-1}$ | $9.25 \times 10^{-2}$ |
| SQ | $3.42 \times 10^4$ | $2.13 \times 10^4$ | $3.24 \times 10^{-1}$ | $2.10 \times 10^{-1}$ | $1.25 \times 10^{-1}$ | $9.10 \times 10^{-2}$ |
| LSQ | $\mathbf{2.90 \times 10^4}$ | $\mathbf{1.12 \times 10^4}$ | $3.04 \times 10^{-1}$ | $1.99 \times 10^{-1}$ | $1.21 \times 10^{-1}$ | $8.57 \times 10^{-2}$ |
| DPQ | $4.01 \times 10^4$ | $2.48 \times 10^4$ | $4.58 \times 10^{-1}$ | $3.54 \times 10^{-1}$ | $1.77 \times 10^{-1}$ | $1.60 \times 10^{-1}$ |
| DPgQ | $3.30 \times 10^4$ | $2.10 \times 10^4$ | $3.29 \times 10^{-1}$ | $2.12 \times 10^{-1}$ | $1.31 \times 10^{-1}$ | $8.74 \times 10^{-2}$ |
| DRQ | $4.75 \times 10^4$ | $2.88 \times 10^4$ | $3.52 \times 10^{-1}$ | $2.54 \times 10^{-1}$ | $1.61 \times 10^{-1}$ | $1.01 \times 10^{-1}$ |
| UNQ | $4.14 \times 10^4$ | $2.33 \times 10^4$ | $3.52 \times 10^{-1}$ | $2.39 \times 10^{-1}$ | $1.48 \times 10^{-1}$ | $1.08 \times 10^{-1}$ |
| Ours | $\underline{2.92 \times 10^4}$ | $\underline{1.91 \times 10^4}$ | $\mathbf{2.92 \times 10^{-1}}$ | $\mathbf{1.93 \times 10^{-1}}$ | $\mathbf{1.02 \times 10^{-1}}$ | $\mathbf{6.72 \times 10^{-2}}$ |

Table 3: Comparisons of quantization error with state-of-the-arts on three datasets (*lower is better*). Ours achieves the lowest quantization error in most cases. This gives us benefits of feature reconstruction. Observe that UNQ performs poorly, we believe it focuses more on ranking and similarity preservation, other than reconstruction.

Ours get the 2nd place on SIFT1M, and the lowest on remaining datasets in most cases. Quantization error indicates reconstruction accuracy and further shows the quality of codebook generation and quantization codes selection. Notably, ours significantly outperforms UNQ, which has a strong bias on the reconstruction task. This is because they focus more on ranking, not the quantization error. The result shows that our method can be applied to other areas, *e.g.* vector compression.

## 5.4 Ablation study

Our ablation study is conducted on the SIFT1M dataset, with the code-length of 32 bits, which in our experiments is sufficient to show how does each component affects our model. We choose the following variants to perform ablation:

| Method | SIFT1M@32 bits | | | |
|---|---|---|---|---|
| | QE | R@1 | R@10 | R@100 |
| w/o regularization | $3.38 \times 10^4$ | 7.60 | 29.96 | 68.73 |
| w/o return-norm | $\underline{3.06 \times 10^4}$ | $\underline{10.57}$ | $\underline{36.44}$ | $\underline{76.04}$ |
| w/o correction | $3.10 \times 10^4$ | 10.09 | 35.30 | 75.16 |
| w/o refinement | $3.17 \times 10^4$ | 9.91 | 30.39 | 68.28 |
| DeepQ | $\mathbf{2.92 \times 10^4}$ | $\mathbf{11.02}$ | $\mathbf{37.73}$ | $\mathbf{76.79}$ |

Table 4: Ablation study conducted on SIFT1M with 32 bits code-length. Entropy regularization forces network to try more codeword combinations, which help to jump out of local-optima. Return normalization and value correction help for fast convergence. The extra refinement leads to low quantization error and high recall with negligible costs.

**w/o regularization**: which removes $e_\theta$ in the losses, and the output distributions will not be forced to be uniform.

**w/o return-norm**: which does not normalize $R$, and therefor advantage is computed by $R$ other than $\bar{R}$.

**w/o correction**: which removes value correction. So our VC-PPO falls back to the original PPO.

**w/o refinement**: which directly encode the base set without extra refinement.

Quantization error and recall are evaluated and placed in Table 4. We report the best value they ever met during the training procedure. Specifically, when regularization is removed, it seems that the network is trapped in local-optima and the performance drops. Meanwhile, although return normalization and value correction give us only subtle improvements, we find they help the network to converge quickly. The extra refinement gives us lower quantization error and higher recall, specially when we want to perform fast training before the network is converged.

## 6 Conclusion and Future Work

In this paper, we first review previous works of constrained MCQs, and investigate solutions to unconstrained ones. Since finding the global-optima of MCQ is NP-hard, researchers apply constraints to find near-optimal solutions or employ heuristic algorithms that are still time-consuming. This paper takes the first attempt to find a *deep* solution to MCQ. The proposed IndepNet is designed to be simple enough to encode vectors extremely fast. Furthermore, our network shows state-of-the-art performance in retrieval and reconstruction tasks. Our method is slow to converge in a large dataset, which hinders our performance. So, our future work will focus on training speedup.

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
