# Supplementary Material: Revisiting Multi-Codebook Quantization

In this supplemental material, we will first give the detailed network implementation for reproducibility (section A). Then, additional experimental results are illustrated in section B for comprehensive analysis.

## A    Model Specifications[1]

In this section, specifications of *IndepNet* $\theta$ and *QENet* $\tau$ are explained, as well as hyper-parameters for reproducibility. The specification on each dataset (SIFT1M, DEEP1M and LabelMe22K) are shown in Table 1. Since two networks only consist of IndepBlocks, we only show the specification of a IndepBlock for simplicity. Additionally, during training, we insert dropout layers after every layer-normalization in all layer-groups to tackle overfitting. Table 2 shows all hyper-parameters involved in our experiments.

| Layer № | SIFT / DEEP ($D = 128/96$) | | LabelMe22K ($D = 512$) | |
|---|---|---|---|---|
| | Out-dim | Layers | Out-dim | Layers |
| 1 | $4D$ | $\lg(D, 4D)$ | $D$ | $\lg(D, D)$ |
| 2 | $2D$ | $\lg(4D, 2D)$ | $D/2$ | $\lg(D, D/2)$ |
| 3 | $D$ | $\lg(2D, D)$ | $D/4$ | $\lg(D/2, D/4)$ |
| 4 | $D$ | $\lg(D, D)$ | $D/4$ | $\lg(D/4, D/4)$ |
| 5 | $2D$ | $\lg(D, 2D)$ | $D/2$ | $\lg(D/4, D/2)$ |
| 6 | $4D$ | $\lg(2D, 4D)$ | $D$ | $\lg(D/2, D)$ |
| 7 | $7D$ | $\mathrm{cat}(4, 5, 6)$ | $7D/4$ | $\mathrm{cat}(4, 5, 6)$ |
| 8 | $K$ | $\mathrm{fc}(7D, K)$ | $K$ | $\mathrm{fc}(7D/4, K)$ |

Table 1: Layer specifications on three datasets. $D$ is the dimension of input $\boldsymbol{x}$. $\mathrm{fc}(\cdot, \cdot)$ is a linear layer with input and output dimensions specified. $\lg(\cdot, \cdot)$ is a layer-group which has a linear layer $\mathrm{fc}(\cdot, \cdot)$ with a ReLU activation and a layer-normalization. $\mathrm{cat}(\cdots)$ concatenates layers with specified №, *e.g.* $\mathrm{cat}(4, 5, 6)$ concatenates the outputs of layer 4, 5 and 6. We also show the output dimensions after each layer in the 2nd and 4th columns. On SIFT1M and DEEP1M, we first expand $\boldsymbol{x}$ to $4D$, while on LabelMe22K, since the dimension of $\boldsymbol{x}$ is enough, we keep the input dimension as $D$.

## B    Additional Experimental Results

**Encoding times.**    The total encoding time w.r.t. code-length comparison is stated in Table 3. Ours is faster than most of constrained MCQs, and much faster than LSQ. Although UNQ is the fastest among all methods, it still needs to decode and re-rank during the retrieval.

---

[1]Our implementation is publicly available at this url.

| Method | SIFT1M total encoding time | | | |
|--------|---------|---------|---------|----------|
|        | 16 bits | 32 bits | 64 bits | 128 bits |
| OPQ | 4.44 | 4.19 | 4.13 | 4.30 |
| SQ | 3.58 | 5.18 | 10.28 | 20.67 |
| LSQ | 52.84 | 96.99 | 256.86 | 639.18 |
| DPQ | 5.60 | 6.57 | 8.81 | 12.81 |
| DPgQ | 6.46 | 8.71 | 13.67 | 24.42 |
| DRQ | 6.42 | 8.88 | 13.78 | 24.55 |
| UNQ | 3.31 | 3.34 | 3.36 | 3.40 |
| Ours* | 4.17 | 5.40 | 7.51 | 14.73 |
| Ours | 4.46 | 5.46 | 8.26 | 16.64 |

Table 3: Total encoding time w.r.t. bit-length on SIFT1M dataset (sec). *Ours\** is the variant that removes extra refinement. This is the quantitative results of the time comparison which is visualized in our main paper. Our two variants show superior encoding efficiency among most of compared methods. Compared to LSQ, ours is $11.8\times$, $17.8\times$, $31.1\times$ and $38.4\times$ faster. UNQ achieves the fastest encoding speed but slows down the retrieval because of the decoding and re-ranking.

**16-bits recalls.** 16-bits recall@$\{1, 10, 100\}$ are shown in Table 4 (2 sub-codebooks, 256 codewords for each). Ours acheives comparable recalls against state-of-the-arts on SIFT1M and DEEP1M, and outperforms state-of-the-arts by $0.80\%$, $2.60\%$, $3.20\%$ on LabelMe22K.

**Ablation study.** We visualize the codewords assignment histogram of 4 variants (**w/o regularization**, **w/o return-norm**, **w/o correction**, **w/o refinement**) with 32 bits, on SIFT1M, shown in Figure 4. The quantization error $E$ during sampling stage for each step is shown in Figure 1 (only first $1,000$ steps of $E$ are plotted due to the space limitation). As shown in two figures, training without regularization is trapped in local-optima, *i.e.* datapoints are assigned to a few specific codewords. Meanwhile, without return normalization and value correction, the network is slow to converge.

**Training statistics.** Quantization error $E$, $\mathcal{L}_\theta$ and $\mathcal{L}_\tau$ during training are illustrated in Figure 3. The final codewords assignment histogram is placed in Figure 2. With our proposal, the network gets improved continuously.

| Hyper-parameters | Values |
|------------------|--------|
| Learning rates $\eta_1$, $\eta_2$ | $2 \times 10^{-4}$ with $decay = 0.9999$ |
| Batch size | 2000 |
| Dropout rate | 0.1 |
| Entropy reg. coeff. $\alpha$ | 0.05 with $decay = 0.9999$ |
| Clip-range $\epsilon$ | 0.2 |
| Gradient clipping | 0.5 |

Table 2: Hyper-parameters we employ on all datasets. $decay$ is the exponential decay which is applied after every updating stage *i.e.* in the end of training loop, $\eta_1$, $\eta_2$ and $\alpha$ are multiplied by $decay$. Furthermore, dropout layers are inserted after every layer-normalization during training to tackle overfitting. Additionally, gradient is clipped by its $l2$ norm when updating the networks to avoid training crash.

| Method | SIFT1M@16 bits | | | DEEP1M@16 bits | | | LabeMe22K@16 bits | | |
|---|---|---|---|---|---|---|---|---|---|
| | R@1 | R@10 | R@100 | R@1 | R@10 | R@100 | R@1 | R@10 | R@100 |
| OPQ | 0.23 | 2.17 | 13.19 | 0.27 | 2.25 | 14.90 | 3.35 | 20.20 | 65.35 |
| SQ | 0.33 | 3.95 | 24.24 | 0.47 | 3.81 | 25.09 | 4.00 | 25.20 | 72.70 |
| LSQ | **0.57** | **4.92** | **28.81** | **0.48** | **4.22** | **26.96** | 4.45 | 27.25 | 75.80 |
| DPQ | 0.31 | 3.28 | 18.45 | 0.13 | 1.41 | 9.41 | 2.25 | 16.50 | 63.00 |
| DPgQ | 0.41 | 4.43 | 26.37 | 0.41 | 3.62 | 24.07 | 4.30 | 26.60 | 74.50 |
| DRQ | 0.38 | 3.58 | 24.96 | 0.35 | 3.48 | 22.64 | 1.00 | 9.15 | 52.20 |
| UNQ | 0.42 | 4.27 | 25.91 | 0.34 | 3.66 | 23.91 | 7.75 | 33.55 | 80.35 |
| Ours | 0.53 | 4.34 | 27.68 | 0.45 | 3.79 | 26.54 | **8.55** | **36.15** | **83.55** |

Table 4: Quantitative comparisons with state-of-the-arts on three datasets with 16 bits (2 sub-codebooks and 256 codewords for each). Recall(R)@$\{1, 10, 100\}$ are reported (%). The 16 bits' results are similar with others that are shown in the paper. Specifically, our method has comparable retrieval performance against the state-of-the-arts on SIFT1M and DEEP1M datasets. Meanwhile, ours outperforms state-of-the-art by $0.80\%$, $2.60\%$, $3.20\%$ on LabelMe22K.

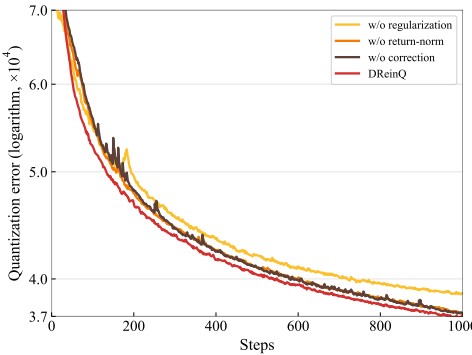

Figure 1: Quantization error during training for $4$ variants. Comparing to other three variants, **w/o regularization** has lower error in the first few steps, but it is quickly trapped into local-optima and hard to converge. **w/o return-norm** and **w/o correction** have higher errors and are slower to converge than the full version during the whole training. For example, ours reaches quantization error of $4.0 \times 10^4$ at ~ $550$ steps while the two variants reach later at ~ $600$ steps.

Figure 2: Codewords assignment histogram for 32 bits on SIFT1M dataset. We visualize the histogram by encoding on the whole base set and generating histogram of quantization codes seperately on 4 sub-codebooks. Higher bars indicate there are more datapoints assign to these codewords.

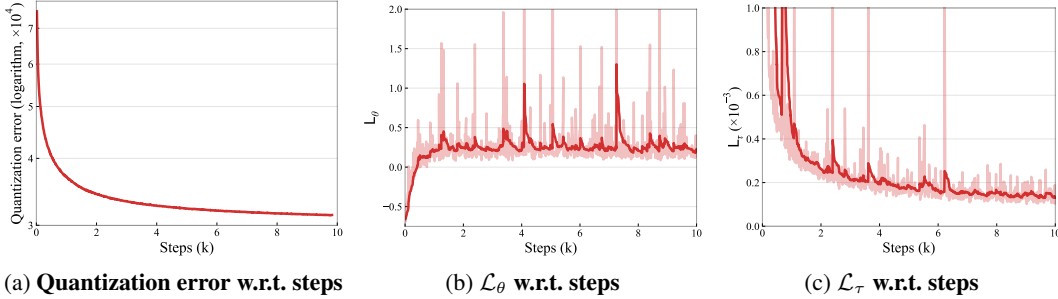

(a) **Quantization error w.r.t. steps**  (b) $\mathcal{L}_\theta$ **w.r.t. steps**  (c) $\mathcal{L}_\tau$ **w.r.t. steps**

Figure 3: Training statistics on SIFT1M dataset, 32 bits. We plot the quantization error of training set after each sampling stage, and $\mathcal{L}_\theta$, $\mathcal{L}_\tau$ during updating stage. The quantization error declines continuously.

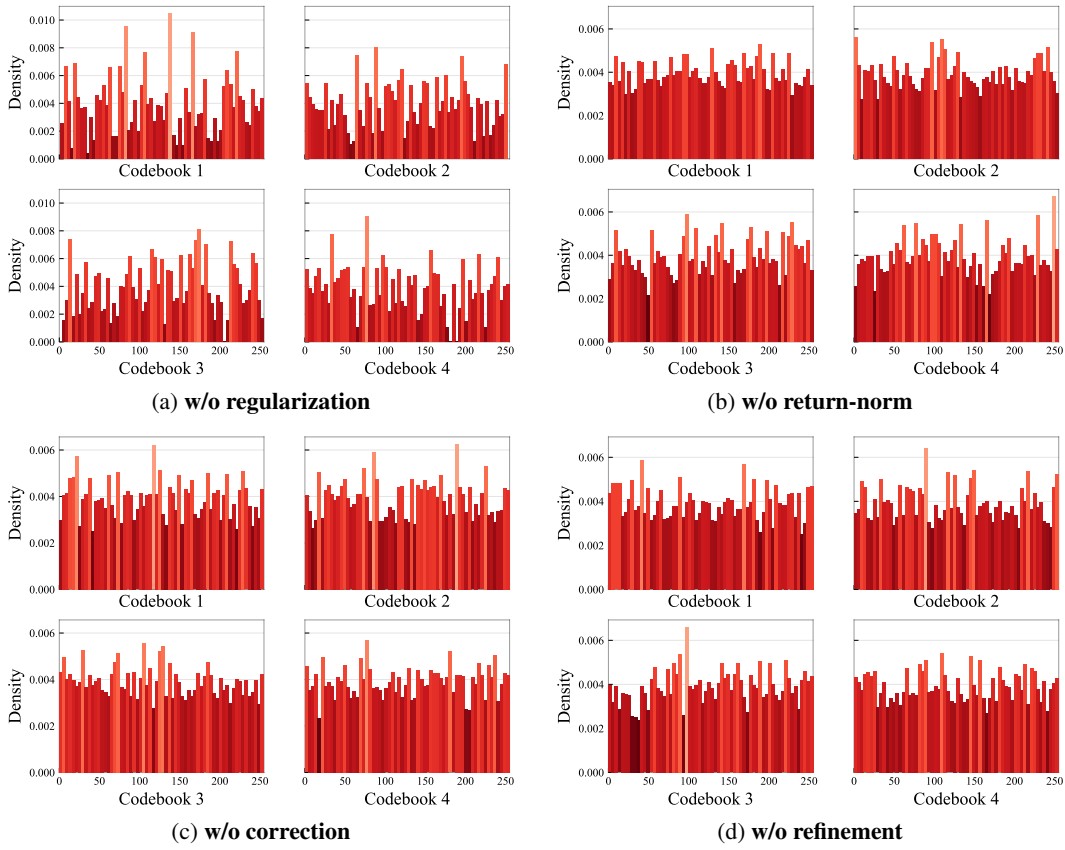

Figure 4: Codewords assignment histogram of 4 variants, 32 bits on SIFT1M. Compared to Figure 2, the variant **w/o regularization** has higher variance on assignment, *i.e.* datapoints are assigned to a few specific codewords while some codewords are totally not used. Other three variants have similar histogram as the full version, but actually have higher quantization error and lower performance which are shown in other experiments.