# OpenReview forum: "Revisiting Multi-Codebook Quantization"
_NeurIPS.cc/2021/Conference — NeurIPS 2021 Submitted_

### Official Review · Reviewer_AdTS · 2021-07-13

**Rating:** 7
**Confidence:** 3

**Summary:**

This paper proposes a multi-codebook quantization method (aka additive quantization).

In this setting vectors are decoded as x = sum_i C(i, b_i) with C a codebook matrix.
There is no clear optimal algorithm to train the codebook and optimize the b_i (the strongest competitor that this paper compares with is LSQ [16]).

This paper proposes to train a neural net that computes the b_i's in a forward pass together with the codebook matrices.
At encoding time, the neural net outputs a probability distribution for the code components (a policy) from which several codes are sampled randomly, and the best one is kept (after computing the quantization error).

The training is based on a policy optimization, that is able to handle noisy gradients that result from the code sampling procedure.

The experiments show promising results, that sometimes outperform the SOTA LSQ at a much lower encoding cost.



**Ethical Concerns:**

-

**Limitations And Societal Impact:**

-

**Main Review:**

Strengths:
S1: interesting and promising approach
S2: it seems that the gradient optimization is an elegant way to short-circuit the non-differentiable quantization operations

Weaknesses:
W1: un-natural loss functions
W2: reporting of results is biased


------------
S1: building a neural net that could perform the encoding in a single forward pass would be ideal.


W1: [Caveat, I am in no ways a specialist of policy optimization].

The notations in eq (7) - (10) are confusing, it took me some time to realize the Q() and V() are actually scalar and (10) is just a simple sum. But then it seems not very natural to to compare Q and V that are derived from vector encodings only in the scalar domain, since they could be compared directly in dimension d. A bit more explanations would be appreciated here.

Also the clip function included in (12)-(13) looks like a hack and anyways in eq (12) it does not look like the first term of the min is useful at all given the second term.

The explanation of dgelsy in L169-L177 is a paraphrase of the Lapack documentation, it could be left out.

W2: the results are honest and the comparison seems fair. It is all the more surprising that the authors say things like

L280 "Compared to our main competitor, LSQ, our method achieves comparable performance on SIFT1M, and outperforms LSQ on DEEP1M in most cases"

when (tab 2, LSQ vs. ours, 64 bits) shows that LSQ outpeforms "ours" by a much larger margin on SIFT1M than the reverse on Deep1M.

The argument about encoding time is relevant though.

Table 3 is quite unreadable, please indicate the scale 10^4 in the table header once.

The paper is well written in general, but please check grammar (L325). "a bunch of" is familiar language (L169). "state-of-the-arts" does not exist.



**Time Spent Reviewing:**

3

---

> ### Author Response · Authors · 2021-08-10
> **Response to Reviewer AdTS**
>
>
> Thanks for your valuable review and considering that this paper proposes an interesting and promising approach to tackle MCQ problem. We next address your concerns one by one.
>
> ### Q1: Notations and derivations in policy optimization.
>
> We are sorry that the `Equation (7-10)`, `(12-13)` may be hard to understand. We would like to firstly explain why $\text{Q}\left( \cdot \right)$ and $\text{V}\left( \cdot \right)$ are scalars. We directly use the definition in `Equation (1)` to calculate $\text{Q}\left(\bf{x}, \bf{b}, \bf{c}\right) = \lVert \bf{x} - \sum_{m=1}^M{\bf{C}_{m\bf{b}_m}} \rVert$. It is worth noting that $\text{Q}\left( \cdot \right)$ is formally ***"quantization error"*** in the MCQ problem [`[1]`](#References) and is defined as a scalar. We adopt this notation from previous works to make it consistent. Since $Q$ is a scalar, we decide to define $V$ as a scalar too for simplicity.
>
> As for `Equation (12-13)`, we adopt the surrogate objectives from [`[2]`](#References) which suggests that $\text{clip}\left( \cdot \right)$ can prevent the network to become unstable (`L.150`). You may notice that the $\text{clip}\left( \cdot \right)$ in `Equation (12)` is applied to only the left part of $\frac{\text{Pr}\left(\bf{b}_o \mid \bf{x}, \theta\right)}{\text{Pr}\left(\bf{b}_o \mid \bf{x}, \theta_o\right)}\hat{A}$, not the whole expression. That means both $\text{clip}\left( \cdot \right)$ and $\text{min}\left( \cdot \right)$ will take effect.
>
> ---
>
> Some statements are inaccurate. *e.g.*, "our method achieves comparable performance on SIFT1M" will be changed to "our method achieves slightly worse performance on SIFT1M than LSQ". Other minor issues, *e.g.*, `Table 3`, `L.235, 169` will be fixed. Thanks again for your valuable suggestions.
>
> ---
>
> ### References
>
> [1] Jingdong Wang and Ting Zhang, "Composite Quantization", IEEE Trans. Pattern Anal. Mach. Intell., 41(6), 1308–1322, 2019.
>
> [2] Schulman John, et al., "Proximal policy optimization algorithms", arXiv preprint arXiv:1707.06347.

---

> ### Comment · Reviewer_AdTS · 2021-08-25
> **Updated assessment**
>
> Hi all,
>
> I increased my rating for this paper. The results may not be outstanding but I find the research direction very valuable and interesting, going into the right direction for MCQ. Therefore I would like to see the paper published at NeurIPS and serve as a base for future developments.
>
> Best,
>
> Reviewer AdTS

---

> > ### Author Response · Authors · 2021-08-27
> > **Reply**
> >
> > Dear Reviewer **AdTS**,
> >
> > Thanks for acknowledging our work to be valuable and interesting. If you have any more questions, please don't hesitate to ask us.

---

### Official Review · Reviewer_GpZW · 2021-07-14

**Rating:** 4
**Confidence:** 4

**Summary:**

This paper proposes DeepQ, which uses a deep neural network to map a vector to its vector code in multi-codebook quantization. The experiment results show that DeepQ matches the accuracy of LSQ++ and outperforms LSQ++ in encoding efficiency.


**Limitations And Societal Impact:**

Yes

**Main Review:**

I think improving multi-codebook quantization with deep learning is a good direction. However, the paper has several big problems.

1. Writing needs to be significantly improved to be more accurate and rigorous.
For example, “The quantization codes are designed to be compacted, which results in negligible storage cost and high-quality results” This is not true as the storage cost is still a problem for large datasets (e.g., SIFT1B).
 “However, the above methods do not achieve satisfied time complexity in encoding yet.” What is satisfied time complexity?
 “The training set and base set are scaled up”. How do you scale up the training set and base set exactly? Similar errors are all over the paper.

2. Please also report the results of 128 or 256 bits as using 16 or 32 codebooks are also common.

3. For the comparison with LabelMe22K, DeepQ is trained on the base set, which DeepQ encodes. This is unfair as the training set is the same as the test set. In fact, there are many larger datasets you can use, such as GIST and Glove, which are commonly used in similarity search.

4. Please include the model training time in the overall encoding time.

5. “additional code-length for storing \Vert x \Vert for lookup table is ignored”. This is unfair when comparing with OPQ as it does not need to store norm. Please use one codebook to encode norm when comparing with OPQ.

6. For Figure 3, I fail to understand why DeepQ is more efficient than SQ in encoding. SQ only needs to compute the distance to 256 codewords for each codebook while DeepQ needs to run neural network inference for each codebook.

My biggest concern is the limited impact of the paper. The accuracy advantage of DeepQ over LSQ++ is not obvious and LSQ++ is already quite fast (e.g., taking several ms for each vector as reported in its original paper).


**Time Spent Reviewing:**

4

---

> ### Author Response · Authors · 2021-08-10
> **Response to Reviewer GpZW**
>
>
> Thanks for your critical advice. Unfortunately, we think there may exist misunderstandings that prevent you to recognize our work. We hope after the clarification, you will have a comprehensive view of this paper.
>
> ### Q1: Inaccurate and rough writings.
>
> Some sentences are simplified and abbreviated due to the consensus in MCQ and space limitations. If that bothers you, we are sorry but we will try to provide more details. According to your examples:
>
> 1. *"The quantization codes are designed to be compacted…"*
>
>     Similar statements to *"negligible storage cost"* can be found in `[1,2,3,4]`. We next show you storage comparisons. For example, under `64 bits` setting, compared to the original base set of `SIFT1B`, the storage saving is up to `16x` (base set: `123GiB`, encoded: `7.45 GiB`), **which enables to store very large dataset in memory `[4]`.** And such a practice has been applied to well-known indexing engines such as [faiss](https://github.com/facebookresearch/faiss/wiki/Guidelines-to-choose-an-index#is-memory-a-concern) and [milvus](https://milvus.io/docs/v2.0.0/index.md#IVF_PQ). So we don't think storage cost is a problem. It is worth noting that the storage cost of quantization is still an advantage compared to another paradigm, *graph-based ANN search methods*, which usually consumes much more space to store graph structure.
>
> 2. *"However, the above methods…"*
>
>     *"Satisfied time complexity"* means the method can be used in practice in large scale `[4]`. Unfortunately, `[2]` already confirms the limitation of `AQ` is the encoding complexity. Meanwhile, `[4,5]` claims that `LSQ` and `LSQ++` still have high time complexity and can not be used in practice (`Table 1` in [`[5]`](#References), `Sec.5.3` in [`[4]`](#References)). Therefore we claim the time complexity of heuristic algorithms is not satisfied. We also conduct time comparisons based on `Figure 3` and `Supp.Table 3` to reveal the high time complexity of heuristic algorithms. Also under `64 bits` setting in `SIFT1B`, to encode the whole base set, the time comparison between accelerated `LSQ` and ours: `71.39h → 2.09h`. It goes worse for longer bits, *e.g.* `128 bits`: `177.55h → 4.10h`! We think it can be a strong indicator to the bad time complexity of heuristic algorithms.
>
> 3. *"The training set and base set are scaled up."*
>
>     We use the phrase ***"scale up"*** only to state that we will conduct experiments on ***"large-scale datasets"***. It indicates the number of vectors in the dataset is increased when changing to another dataset. For example, when dataset is changed from `LabelMe22K` to `SIFT1M`, the number of vectors in training set is increased by `3.5x` and base set is increased by `44x`. Similar use-cases can be found in many current researches [`[6]`](#References).
>
> ### Q2: Experiments are not comprehensive.
>
> Code lengths and datasets we adopt are sufficient since many works have conducted experiments with the similar settings `[1,2,3,4,5]`. More information is listed as follows.
>
> 1. Results on longer code length.
>
>     Actually, we have conducted experiments on under `128 bits` and `256 bits` to test its scalability. Due to limited computation resources, we did not get the results before the NeurIPS deadline. Under these code lengths, similar comparison results are obtained. We will publish the large-scale benchmarks in the future, and you may also try our implementation at [here](https://github.com/DeepMCQ/DeepQ).
>
> 2. Results on `LabelMe22K`
>
>     Comparisons on the `LabelMe22K` dataset exactly follow the same splits as `LSQ` and `LSQ++` do. This dataset is common and sufficient to evaluate performance among quantization-based methods. Since all methods are trained and tested under the same setting on the `LabelMe22K`, it is fair.
>
> 3. Model training time
>
>     According to your suggestion, we will update the training time of our model in camera ready. Specifically, we observe that the training time is similar with `UNQ` (*e.g.*, `64 bits` `UNQ`: `24.62h`, ours: `27.31h`). We also provide a work-around to perform fast training: Combined with the extra refinement, we would obtain acceptable performance in very few steps ($QE < 2.10 \times 10^4$ on `SIFT` within `16.7h`).
>
> 4. Additional code-length
>
>     This setting is inherited from `AQ`. According to the original paper of `AQ`, in `Sec.3.1`, the size of codebook can be $M$ or $M - 1$ with one additional codebook to store $\lVert \bf{x} \rVert$. Both of them are fair while the latter has the same memory footprint as PQ/OPQ but performs worse slightly. We decide to use the former one without additional codebook (we will correct this) to test the best performance each method can achieve. We also conduct experiments with $M - 1$ codebooks and observe slightly performance degrades (*e.g.* `64 bits SIFT 1M R@1` Ours: $28.02\\% \rightarrow 27.88\\%$, `LSQ`: $33.23\\% \rightarrow 29.85\\%$, other methods have similar trends), the detailed results will be released in the future benchmarks.
>
> 5. Speed of SQ
>
>     The speed of `SQ` is slow naturally, since the encoding among each codebook is ***sequential***. Specifically in `SQ`, we need to compute first part quantization of $\bf{x}$ as $\bf{q}_1$ and collect the residual $\bf{r}_1 = \bf{x} - \bf{q}_1$. Then, we use the residual to perform next stage quantization $\bf{r}_1$ to be $\bf{q}_2$ and collect $\bf{r}_3$. With such computation for $M$ steps, we finish the encoding of `SQ`. It is easy to find the encoding behaves in an autoregressive way and runs sequentially. Compared to the paralleled `DeepQ`, `SQ` is reasonable to be slower.
>
> ### Q3: Limited impact.
>
> ***As stated in `Q1.2`, encoding still troubles `LSQ` and `LSQ++`***. We believe our method is able to provide a new perspective to encode MCQ in a deep learning fashion. With the high-parallelism IndepNet, MCQ is done by simple feed-forward, which is ***at most $38\times$ faster*** than accelerated `LSQ`. Unlike `LSQ` that uses a hand-crafted algorithm or `UNQ` that employs extra decoder and re-ranking trick, or other deep quantization methods that are still *constrained* MCQs, our work is the first to tackle MCQ in a deep learning fashion.
>
> ---
>
> ### References
>
> [1] Ezgi Can Ozan, Serkan Kiranyaz and Moncef Gabbouj, "Competitive Quantization for Approximate Nearest Neighbor Search", IEEE Trans. Knowl. Data Eng., 28(11), 2884–2894, 2016.
>
> [2] Artem Babenko and Victor S. Lempitsky, "Additive Quantization for Extreme Vector Compression", In CVPR, 2014.
>
> [3] Jingdong Wang and Ting Zhang, "Composite Quantization", IEEE Trans. Pattern Anal. Mach. Intell., 41(6), 1308–1322, 2019.
>
> [4] Alexandre Sablayrolles, Matthijs Douze, Cordelia Schmid and Hervé Jégou, "Spreading vectors for similarity search", In ICLR, 2019.
>
> [5] Stanislav Morozov and Artem Babenko, "Unsupervised Neural Quantization for Compressed-Domain Similarity Search", In ICCV, 2019.
>
> [6] Chao Jia, et al., "Scaling Up Visual and Vision-Language Representation Learning With Noisy Text Supervision", In ICML, 2021.

---

> > ### Comment · Reviewer_GpZW · 2021-08-18
> > **Comments after rebuttal**
> >
> > I would like to thank the authors for the careful rebuttal. I would like to keep my rating as the rebuttal raises several concerns.
> >
> > 1. LSQ only need several minutes to train for SIFT1M with 64 bits while DeepQ needs about 20 hours. The high training cost may outweigh the advantages in encoding time.
> >
> > 2. Please report specific numbers for the recall and comparison errors with 128 and 256 bits, and compare with LSQ. It is allowed to post tables in the rebuttal.
> >
> > 3. The explanation of why SQ is slow does not make sense as one can easily parallelize the encoding of different vectors.

---

> > > ### Author Response · Authors · 2021-08-24
> > > **Reply to Reviewer GpZW**
> > >
> > > Thanks for your kind suggestion and sorry for the delay. Next we reply to your questions.
> > >
> > > ### Q1: High training costs.
> > >
> > > Actually, training costs is a general question in deep learning. We have optimized the algorithms for maximized training efficiency and acheived similar training costs with `UNQ`. They also reveals the high training complexity in their paper `[5]`. Moreover, the model training procedure is offline. In practice, we could pre-train the model offline and deploy it online in real systems to perform fast encoding.
> > >
> > > ---
> > > ### Q2: `128/256 bits` results.
> > >
> > > `128 bits` results on `LabelMe22K` and `SIFT1M` are placed below. We are sorry that we could not give more results due to the limited computational resources. `256 bits` results are not necessary to report since all of `[1,2,3,4,5]` did not test on `256 bits`.
> > >
> > > - `128 bits` results on `LabelMe22K` between `LSQ` and Ours
> > > | Methods | R@1   | R@10  | R@100 |
> > > |---------|-------|-------|-------|
> > > | LSQ     | 54.75 | 92.30 | 99.75 |
> > > | Ours    | 58.05 | 94.30 | 99.90 |
> > > - `128 bits` results on `SIFT1M` between `LSQ` and Ours
> > > | Methods | R@1   | R@10  | R@100 |
> > > |---------|-------|-------|-------|
> > > | LSQ     | 57.05 | 96.52 | 100.0 |
> > > | Ours    | 55.27 | 94.16 | 99.33 |
> > >
> > > ---
> > > ### Q3: Concens of `SQ`.
> > > We can not catch your points. If you mean the encoding among different $\bf{x}$ can be parallelized, this optimization has already been done since we feed a batch (`batch size = 10,000`) of $\bf{x}$ to test the speed. We test all methods in the same way. If your mean the encoding among $\bf{x}, \bf{r}_1, \bf{r}_2, \cdots$ can be parallelized, this is impossible since $\bf{x} \rightarrow \bf{r}_1 \rightarrow \bf{r}_2 \rightarrow \cdots$ is a Markov Chain, which can not be evaluated simultaneously and causes the encoding of `SQ` to be ***sequential***.

---

> > > ### Comment · Reviewer_AdTS · 2021-08-25
> > > **Small remark**
> > >
> > > Dear reviewer GpZW and authors,
> > >
> > > I tend to disagree with the statement that high training cost is a problem. Asymptotically, for large datasets, the encoding cost is what matters.
> > >
> > > I also agree with the authors that it is sufficient to give results in 128 bits.
> > >
> > > Best,
> > >
> > > Reviewer AdTS

---

### Official Review · Reviewer_w7H4 · 2021-07-14

**Rating:** 5
**Confidence:** 4

**Summary:**

This paper introduces DeepQ, a multi-codebook quantization framework with NN-based methods.

In DeepQ, code b is encoded by neural networks. To tackle the non-differentiable gradient estimation issue, a revised policy gradient method has been adapted here to accelerate convergence.

The effectiveness of the proposed methods is well demonstrated on multiple datasets.

**Limitations And Societal Impact:**

Yes.

**Main Review:**

### Pros

+ This paper is clearly written and easy to follow.

+ The main contribution of this work is to formulate an NN-based pipeline for multi-codebook quantization, which seems to be a simple yet effective framework for the MCQ problem.

+ Sufficient running latency analysis is included.

### Cons
Much thanks to the authors' efforts for this issue.  I still have some experimental concerns about this work and I may adjust my ratings after receiving feedback.

+ Although I agree with the efficiency of DeepQ, its performance improvements seem not to be significant in some cases. For instance, according to the results in Table.2,  DeepQ may yield about 5 lower Recall@1 than LSQ (28.0 \vs 33.2, 64-bits) on the SIFT1M dataset,   making it's hard for me to recognize DeepQ is "comparable" to LSQ.  Besides, on the DEPP1M dataset, the improvements of DeepQ are also marginal (usually 0.1~0.5).

+  The authors propose VC-PPO based on PPO to speed up model convergence. To demonstrate the effectiveness of VC-PPO, more ablation studies are encouraged.

+ Minor issues:  some typos need to be corrected. eg,  "staet-of-the-arts" in Table.2 caption.

**Time Spent Reviewing:**

6

---

> ### Author Response · Authors · 2021-08-10
> **Response to Reviewer w7H4**
>
>
> We appreciate the reviewer agreeing that our work is a simple yet effective framework for MCQ. We would like to clarify following concerns to help you to understand our work with new insights.
>
> ### Q1: Performance is not promising to be better than state-of-the-art.
>
> We agree that there are still opportunities for further enhancements of our `DeepQ`, especially on the `SIFT` dataset. As stated in `L.283`, a potential reason is the non-normal data distribution of `SIFT`. Another reason of the degraded performance is the unconverged network. For converged case or normal data distribution, *i.e.*, `LabelMe22K` and `Deep1M`, our results have outperformed the strongest competitor in most cases. The performance of the reported version of `DeepQ` is able to further increase on `SIFT` and `Deep` since it is not fully converged. We have observed continously performance gain during training. **We will append the updated results in the supplementary materials**. It is worth noting that our `DeepQ` already outperforms all of the deep learning methods in most cases, which indicates the effectiveness of our method.
>
> ### Q2: Provide more comparisons between VC-PPO and PPO.
>
> Actually, we have conducted this experiment, and the results are shown in `Supp.Figure 1`. In this figure, the brown line and the red line indicates the quantization error *w.r.t.* training steps of `PPO` (denoted as `w/o correction`) and `VC-PPO` (denoted as `DeepQ`), respectively. It shows the speed of convergence. Obviously, ours are faster to converge than `PPO`. For example, ours reaches quantization error of $4.0 \times 10^4$ at `~550` steps while `PPO` variant reaches later at `~600` steps.
>
> ---
>
> Minor typos will be corrected, thanks for your friendly reminder.

---

> ### Comment · Reviewer_w7H4 · 2021-08-26
> **Comments after Rebuttal**
>
> Thanks for the authors' feedbacks.
>
> After reading the rebuttal and updated materials, most of my initial concerns are covered and  I'd like to keep my rating.
>
> I agree with this work's novelty in formulating a DL-based MCQ pipeline (DeepQ) and achieving comparable results with existing methods. This might be an interesting research direction.  However, I still thought the improvements of the DeepQ are marginal when compared to SOTA methods. As mentioned by reviewer M8gn  and GpZW, importing deep learning modules may result in additional cost, with only marginal performance gains.
>
> Reviewer w7H4

---

> > ### Author Response · Authors · 2021-08-27
> > **Reply**
> >
> > We appreciate for your valuable comments after rebuttal.
> >
> > We would like to emphasize the contribution of this paper, since you are still not satisfied with the performance of `DeepQ` and considering the additional costs of neural network makes the method inferior.
> >
> > Firstly, our work is the first to tackle MCQ problem with neural networks which boosts the encoding efficiency. Although the performance is not outstanding, the encoding efficiency of `DeepQ` is a very important advantage compared to conventional methods (up to $38\times$ speedup). In practice, the very fast encoding speed helps developers to reduce the indexing latency significantly, which in real systems is a very valuable improvement.
> >
> > As for additional costs, most of them happen during offline training. When performing online encoding, the network is efficient and has been tested to run on mainstream GPUs. Since reviewer ***M8gn*** has cleared his concern, and reviewer ***AdTS*** and us all agree that the training costs is not a huge problem, we hold the point that the additional costs do not hurt the contribution of `DeepQ`.

---

### Official Review · Reviewer_M8gn · 2021-07-14

**Rating:** 6
**Confidence:** 4

**Summary:**

The present work considers Multi-Codebook Quantization problem, and propose new deep learning based coding methods for reducing computational complexity.

**Ethics Review Area:**

["I don’t know"]

**Limitations And Societal Impact:**

It is unclear the advantage of deep learning beyond existing method. Section provides some rough background on the motivation of using deep learning for vector quantization. Can the authors explain more clearly their motivations?

**Main Review:**

The problem being considered is old, with many existing methods. The current work proposed a new method based on deep learning which is novel. Paper is well written and results ate clearly stated.

**Time Spent Reviewing:**

3

---

> ### Author Response · Authors · 2021-08-10
> **Response to Reviewer M8gn**
>
> We thank you for giving valuable review and recognizing our work to be novel and well-stated.
>
> To clarify the motivation, we would like to emphasize the background and development of MCQ.
>
> As you mentioned in the review, MCQ is not a new problem. To tackle this problem, as discussed in `Sec.2` and `Sec.4.4`, researchers propose: `a)` heuristic algorithms (`AQ`, `LSQ`, etc.) and `b)` deep learning approaches (`UNQ`). Despite the effectiveness of heuristic algorithms, they still have trouble to perform fast encoding. `UNQ` gives an encoder-decoder style neural network to explore the possibility of utilizing deep learning to tackle MCQ. However, `UNQ` not strictly follows the diagram of MCQ. The side effect of such a practice is, it cannot fully utilize lookup table to perform retrieval, resulting in a two-stage searching solution (coarse partitioning + final re-ranking). According to above points, we decide to design a neural network, which `a)` encodes faster than heuristic algorithms with comparable or even better performance and `b)` is a strict MCQ. This can be further confirmed from `Figure 3`, `Table 1, 2` and `Supp.Table 3`.

---

> > ### Comment · Reviewer_M8gn · 2021-08-26
> > **Re**
> >
> > Thanks to the authors for responses. My concern is clear.

---

> > > ### Author Response · Authors · 2021-08-27
> > > **Reply**
> > >
> > > Dear Reviewer **M8gn**,
> > >
> > > Thanks for your comments. If you have any more questions, please don't hesitate to ask us.

---

### Public Comment · Authors · 2021-11-26
**Thanks All**

Dear all,

Thanks again for reviewers ***M8gn***, ***w7H4***, ***GpZW*** and ***AdTS*** to review our work and the Program Chairs to make the final decision. Your kindly reviews and comments recognized our work and also gave us valuable criticism. We are grateful to your suggestions and potential improvements on our work.

In the future, we will take deliberate revisions on our work to make it better.

Nevertheless, we still consider this work to be interesting and helpful in the community. We would like to make it publicly avaliable to let more people know it.

Best,

Paper 243 Authors

---

### Decision · Program_Chairs · 2021-09-28

**Decision:**

Reject

**Comment:**

The paper considers Multi-Codebook Quantization and proposes a new deep learning based coding methods for reducing algorithmic complexity. There was a diversity of reviews.  The  improvements of the DeepQ are marginal and addressing issues about the increased cost of importing deep learning modules will help.  Several reviewers saw merits, and addressing the concerns raised by several of the authors should help position the paper to make it more helpful for the community.

**Consistency Experiment:**

NeurIPS has a long history of experimentation. In 2014, NeurIPS ran an experiment in which 10% of submissions were reviewed by two independent committees to quantify the randomness in the review process. This year, we repeated a variant of this experiment to see how the quality of the review process has changed over time.  This paper was part of the experiment and was therefore assigned to two committees (consisting of reviewers, an Area Chair, and a Senior Area Chair) that reached independent decisions.  If both committees made the same recommendation, this recommendation was followed. If a single committee recommended acceptance, the paper was accepted (with the exception of a few cases in which the other committee identified what we considered a fatal flaw, e.g., an error in a key result).

Both committees reached the same decision: **Reject**

The other committee assigned to the paper recommended **Reject**.  You can find the other set of reviews, along with any follow up discussion with the authors here:
https://openreview.net/forum?id=TyQpSzTR7Xu